# Solar Radiation Flux Provides a Method of Quantifying Weed-Crop Balance in Present and Future Climates

**DOI:** 10.3390/plants10122657

**Published:** 2021-12-03

**Authors:** Geoffrey R. Squire, Mark W. Young, Cathy Hawes

**Affiliations:** James Hutton Institute, Dundee DD2 5DA, UK; mark.young@hutton.ac.uk (M.W.Y.); cathy.hawes@hutton.ac.uk (C.H.)

**Keywords:** weeds, crop growth, solar flux, radiation, plant diversity, climate, adaptation

## Abstract

A systematic approach to quantifying the weed–crop balance through the flux of solar radiation was developed and tested on commercial fields in a long-established Atlantic zone cropland. Measuring and modelling solar energy flux in crop stands has become standard practice in analysis and comparison of crop growth and yield across regions, species and years. In a similar manner, the partitioning of incoming radiation between crops and the in-field plant community may provide ‘common currencies’ through which to quantify positive and negative effects of weeds in relation to global change. Here, possibilities were explored for converting simple ground-cover measures in commercial fields of winter and spring oilseed rape in eastern Scotland, UK to metrics of solar flux. Solar radiation intercepted by the crops ranged with season and sowing delay from 129 to 1975 MJ m^−2^ (15-fold). Radiation transmitted through the crop, together with local weed management, resulted in a 70-fold range of weed intercepted radiation (14.2 to 963 MJ m^−2^), which in turn explained 93% of the corresponding between-site variation in weed dry mass (6.36 to 459 g m^−2^). Transmitted radiation explained almost 90% of the variation in number of weed species per field (12 to 40). The conversion of intercepted radiation to weed dry matter was far less variable at a mean of 0.74 g MJ^−1^ at both winter and spring sites. The primary cause of variation was an interaction between the temperature at sowing and the annual wave of incoming solar radiation. The high degree of explanatory power in solar flux indicates its potential use as an initial predictor and subsequent monitoring tool in the face of future change in climate and cropping intensity.

## 1. Introduction

Agricultural fields contain a cultivated crop or pasture and a community of wild plants or weeds that exists from year to year through a soil seedbank [1,2,3]. The seedbank consists of species that are detrimental to production in that they take resource from the crop, or contaminate harvest and species that are beneficial in supporting ecological functions such as decomposition, pollination and biological pest control [4,5,6,7]. Sustainable production requires that both economic output and beneficial supporting functions are maintained [8,9,10], yet achieving such a balance [10] is likely to become more problematic given predicted changes in climate [11,12,13]. The future balance will be influenced by the relative responses of weeds and crops to factors including atmospheric carbon dioxide concentration and temperature, operating through phenology, growth and dispersal [12]. Weeds adapt and evolve across wide spatial and temporal scales, including distribution or range at the landscape scale, niche in the arable community, and the traits that determine interactions between individuals and their resource [13]. Given the complexity and uncertainty of future weed shifts [11,12], improved systems of monitoring and forecasting are essential to devise mitigation and management strategies for minimising crop loss while maintaining ecological function [14].

In preparation for an uncertain future, therefore, weed science and practice need to command a range of management options and methodologies for monitoring and predicting change in the weed–crop balance. The capacity of management to adapt to weed shifts has been restricted in recent decades by a technological lock-in to chemical pesticides and a cultural resistance to adopt new approaches [15,16]. However, an increasing number of non-chemical interventions are being developed, tested and put into practice [17]. Advances in modelling that link trait, niche and distribution range [14] to economic output and ecological function are enabling better prediction and forecasting [16,18,19]. Yet rapid, practical in-field methods are still essential for quantifying weeds and crops, for example, in extensive trialling or monitoring programmes that need to cover hundreds of field sites over several years [20,21].

This paper explores a means of quantifying partition of solar radiation in weed–crop stands through a simple methodology based on field assessments in a region of arable cropland in the east of Scotland UK. The approach is adapted from well-established concepts in crop physiology and modelling. Notably, the use of resource-flux as a ‘common currency’ [22,23] initiated a step-change in understanding the links between crop production and climatic variables. Expressing carbon acquisition through solar radiation, and also water and nitrogen, allowed crops to be quantified by a few core attributes such as cumulative resource capture and use-efficiency [24,25]. Direct comparisons became possible over the range of annual incoming solar between tropical and north-temperate regions, among monsoonal and semi-arid climates, and between annuals and perennials, monocultures and intercrops [25,26]. Flux-based methods have been widely applied in global modelling platforms for predicting carbon uptake and storage in vegetation and soil over large tracts of the Earth’s surface [27]. 

The approach taken here is to derive solar radiation flux from measures of ground cover by crops and weeds. Although methods of converting percentage crop cover to solar flux have been in use for many years [28], the analysis combines crop and weed cover in a common approach. As in most agricultural regions, weeds were historically a major limitation, causing frequent crop failure that resulted in Acts of Parliament as early as the 14th and 15th centuries intended to control damaging species such as *Chrysanthemum segetum* [29]. Weeds remained a major challenge to food security in the early 20th century, through competition with crops and the unwitting harvest of poisonous seed or foliage [30], to a degree that five weed species, including *Senecio jacobaea*, were named in a 1959 Act of Parliament aimed at control and eradication [31]. Subsequently, chemical herbicides and more competitive crop varieties have suppressed weeds to the point where most of the damaging species are well regulated [3]. Weeds nevertheless remain a potential problem through their long-lived seedbank, which can rise to detrimental levels under lax or ineffective management [32]. Although to date there has been negligible study of the effects of climate on weeds in the region, the prediction to the year 2050 of slight warming coupled with drier summers, wetter winters and more extreme weather [33] gives cause for concern, shared globally [11,12,13,14], that the weed–crop balance may again be disrupted. 

The future fate of weeds in broadleaf ‘break crops’ that are inserted in an otherwise cereal-based system is of particular interest. The break crops allow control of economically important grass weeds and, at the same time, allow a richer weed flora to support the arable food web [6,8,9,10,34]. Traditionally, break crops such as oilseed rape comprised both short-season (spring sown) and long-season (autumn sown) varieties, but following intensification between 1960 and 2000, oilseed rape has become exclusively long-season. However, the crop has been susceptible to yield-depression in some recent cold, wet winters, a problem that is likely to worsen under future predicted climates. A possible return to short-season breaks may therefore be necessary, but the effects of such a transition, and the consequences of climatic change generally, on yield, pesticide use, weed competition and weed floral diversity are highly uncertain.

The data used here are of particular value to the questions in hand because they were collected at a time (2000–2004) when spring- and autumn-sown varieties were both grown commercially. Monitored fields of each of spring and winter types are known from prior examination to present diverse combinations of crop and weed cover [35], likely due to a ‘time of sowing’ effect in which small delays after an optimum time lead to reduced crop growth [36], and hence increased opportunity for weed growth. The specific aims of the study were to (1) identify the metrics of solar flux that have the greatest influence on attributes of the weed layer, (2) define the phases in crop growth that had the greatest influence on weed species and mass, and (3) assess whether the methodology would be sensitive enough to monitor and evaluate the weed–crop balance in response to change in climate and a return to less intensive short-season cropping. The potential of the approach to examine crop–weed trade-offs more widely is considered. 

## 2. Materials and Methods

### 2.1. Sites, Crop Season and Weather Data

To establish and test the methodology proposed here required a set of field sites that differed in canopy development of crop and weed, thereby leading to wide variation in intercepted and transmitted solar radiation flux, but differed as little as possible in potentially confounding effects of latitudinal variation in solar income and related climatic factors. To satisfy these needs, farm fields within a discrete region were selected from among a UK-wide set of experimental sites sown with winter or spring oilseed rape, *B**rassica napus* L. [37]. All fields that occurred between latitudes 56° and 58° N in the East Scotland region were examined: 5 fields sown in 2000, 7 in 2001 and 3 in 2002. The area is within the Atlantic climatic zone of western Europe, characterised by an oceanic climate and moderate annual rainfall, over which solar energy income varies by only a few percent, as defined in the European Solar Radiation Atlas [38]. The daily incoming solar radiation (S) was measured at a meteorological site (latitude 56.46° N; longitude 2.97° W) by a Kipp solarimeter sensitive to global short-wave radiation in the wavelength range 300 to 2800 nm. A typical annual curve, represented by a sine wave fitted to daily measurements, is shown Figure 1. The annual total solar income averaged 3.3 GJ m^−2^ over the years of the experiment. Air temperature is highest in August, the regional mean over the years of monitoring being 13.94 °C, and lowest in January, regional mean 2.56 °C [39]. The months of December, January and February generally experience several days when the temperature is below zero. Standard temperature recordings at the same met site as measured solar radiation (approximating daily mean temperature) illustrate the annual temperature cycle, which lags 30–50 days behind the solar cycle (Figure 1). 

Sites comprised 6 fields of spring varieties (SR) sown in March or April to be harvested in August or September, and 9 fields of winter varieties (WR) sown in August or September to be harvested the next year in July or August. Preliminary examination showed that the set of fields included the wide range of crop–weed combinations previously defined by time-series analysis [35]. All fields were originally subject to a split-field experimental design, but only the half of each field grown with the variety adapted to the region and local weed management was analysed here. Metazachlor (Butisan) herbicide, intended to control a broad spectrum of weed species, was applied once, just before sowing or emergence of the crop at 14 of the 16 sites [37]. At one of the other sites, a spring crop, no herbicide was applied; at the other, a winter crop, herbicide was applied late in the year, months after sowing. At harvest, most crops were desiccated by herbicide or swathed (cut and left to dry in the field) in August or September, WR before SR (Figure 1). Agronomic inputs for each of WR and SR were summarised from standard government surveys for fertiliser [40] and pesticide [41]. 

### 2.2. Methodology for Estimating Intercepted and Transmitted Solar Radiation from Ground Cover

Ground cover had been estimated in the original experiments for crops and weeds about every 14 d from sowing for spring oilseed rape and 28 d for winter oilseed rape. Cover, on a scale of 0–1, was used to derive measures of solar radiation flux. First, the fraction of the incoming solar radiation intercepted by the crop on any day was assumed equivalent to its fractional ground cover (fc), as demonstrated for a range of temperate crops [28]. This fraction was interpolated linearly between sample times and applied to the incoming solar radiation (S) to give a daily intercepted radiation for the crop, Si-c (Equation (1)):Si-c = S·fc (1)

Values were either presented as a daily mean over a stated period (units: MJ m^−2^ d^−1^) or accumulated to give a total over a stated period (units: MJ m^−2^). Representative time courses of Si-c for early sown winter and spring crops are superimposed on the annual curves of solar radiation and temperature in Figure 1. Winter crops show an initial autumn rise, near-levelling over the winter, then a rapid rise in spring beginning on day 50–100. Spring crops show a rapid rise from sowing around day 100, then a slower rise after day 200 with the late summer decline in incoming radiation.

Transmitted radiation was estimated daily as incoming minus intercepted radiation (Equation (2)):St = S − Si-c (2)

Following the procedure applied to the crop, the fraction of radiation intercepted by the weeds on any day was assumed equivalent to their fractional ground cover (fw) from which radiation intercepted by weeds was estimated from Equation (3): Si-w = St.fw (3)

As for Si-c, Sf and Si-w were presented as either a daily mean (units: MJ m^−2^ d^−1^) or accumulated (units: MJ m^−2^). The conversion coefficient, or solar radiation use efficiency (Es), for weeds was calculated as weed dry matter (Wm) divided by cumulative Si-w, as in Equation (4):Es = Wm/Si-w (4)

### 2.3. Weed Dry Matter and Species

Crop dry matter was not measured systematically in the original experiment, because of cost and an emphasis on biodiversity. Weeds were sampled for dry matter and species composition typically in late July and August in the month prior to harvest, at 24 sample points comprising 2 points on each of 12 transects running into the field. Prior measurements showed that weed mass and emerged species were not likely to be limited by the seedbank at these sites. Soil seedbanks in the region typically have seed densities between 2000 and 10,000 seeds per square metre of field, contain typically as many as 40 species, mostly broadleaf but also grass, and lie towards the higher end of the range of seed density and species number for the UK [3,32]. Weeds were sorted into individual taxa, usually to species except where they were too small to identify, in which case they were grouped to genus. Weeds were dried in an oven at 70 °C and their mass presented per unit area (g m^−2^). Of 79 taxa found at the 16 sites, all but two tree species (in very small numbers) were typical of the arable flora. There were 64 taxa at spring crop sites combined and 57 at winter crop sites; 39 taxa were common to both types of crop. A group of common species constituted most biomass. Those present at all or all but one site were *Capsella bursa-pastoris*, *Matricaria* species, *Myosotis arvensis*, *Poa annua*, *Polygonum aviculare*, *Stellaria media* and *Viola arvensis*. Other species present at more than half the sites included *Fallopia convolvulus*, *Fumaria officinalis*, *Galeopsis* species, and *Persicaria maculosa*. Although these common species can emerge throughout the year, most of them have a preference for spring germination. At most winter crop sites, negligible or small weed cover developed in autumn. In both winter and spring crops therefore, weed growth began predominantly in spring, earlier for winter than spring crops (as described later). The Shannon index was calculated as −sum(p.logp) for each site based on the mass of species (and where p is the fraction of each) and presented as an index from 0 to 1 [42]. Some species were present in very small quantities, for example, as a few seedlings identified by their cotyledons, and so a further measure was derived as the number of species having a mean mass >0.1 g m^−2^, an arbitrary but realistic limit signifying the establishment of plants beyond the seedling phase and capable of reproduction. 

### 2.4. Analyses

The effects of agronomy (mainly sowing date) and climate on the solar radiation intercepted (Si-c) and transmitted (St) by the crop were examined first. Transmitted radiation was then used as an independent variable to assess weed attributes: intercepted radiation (Si-w), weed dry matter and weed species number (both total and those with >0.1 g m^−2^ dry matter). Relations between variables were assessed by generalised linear regression based on a normal distribution using the Genstat statistical package (18th edition, VSN International 2015). Regression equations are presented with the associated F-probability (F. pr.) and percentage variation accounted for (%var). 

### 2.5. Interpreting Effects of Sowing Time through Temperature and Solar Radiation

The downstream effects of sowing date acting through the annual cycles of temperature and solar radiation are interpreted through an equation defining fractional ground cover (fc = fraction of incoming radiation intercepted) in terms of a time–temperature integral, generally referred to as thermal time [43]. The relation between fc and thermal time is defined in Equation (5) [44], which has been used previously to interpret and model growth and resource use by crops in this region [45]:fc = *a*/[1 + exp{−*b*(*Ø* − *m*)}](5)
where *Ø* is a time–temperature integral above a base temperature and *a*, *b* and *m* are parameters of the S-shaped curve defining the time course of fc: *a* being the maximum value of fc attained, *m* the approximate mid-point of the curve, and *b* the relative steepness of the S-shape. The time–temperature integral was estimated from summed daily values of mean air temperature above a base temperature at which development does not proceed. The equation is used here to show how small differences in temperature subsequently influence Si-c and St and hence potentially affect Si-w. The equation is fitted to measured values of fc at the first-sown winter oilseed rape site (that had the highest Si-c) by setting parameter *a* to the maximum measured fc and manipulating parameters *b* and *m* to fit the values of fc before winter. The thermal duration from sowing to maturity of the earliest crop was 1490 °C d^−1^. Temperature (e.g., a rise by 1 °C) or time of sowing are then altered and the resulting time courses of fc, Si-c and St compared. 

## 3. Results

### 3.1. Time and Temperature Effects on Crop Solar Radiation Flux

Daily incoming solar radiation (S) and fractional vegetation cover were used to derive the three metrics of solar flux for each field, calculated daily: solar radiation intercepted by the crop (Si-c), that transmitted by the crop (St) and that intercepted by the weed layer (Si-w). Although incoming solar radiation determined the maximum flux on any day, temperature and agronomic factors affected the actual fluxes among crops and weeds through strong effects on the rates of emergence and development [45].

The earliest winter crop was sown on day −137 (relative to 1 January) when the annual temperature wave was just over the maximum. All other crops were sown later, experiencing a decline in temperature (Figure 2a), which, quantified as the mean over 30 days after sowing, was reduced from 14.1 °C for the earliest to 11.6 °C for the latest sown, a downward shift of −2.5 °C or −0.067 °C d^−1^. The earliest sown crop emerged and began early growth when the solar income was still relatively high, receiving a total of 700 MJ m^−2^ up to the beginning of December (Figure 2b). Later crops emerged in steeply declining solar income, receiving 11.7 MJ m^−2^ less for each day of delay after the first sown, the last sown receiving only 400 MJ m^−2^. All winter crops then received a similar solar income up to maturity the following year (not shown). Due to the combination of meteorological conditions summarised in Figure 2, the early winter crops produced greater ground cover before winter than late sown crops. The sparser canopies forming in late-sown crops were more likely to be damaged over winter, so they were less able to expand and intercept solar radiation when conditions ameliorated in the spring and Si-c began to increase rapidly. The overall effects of delayed sowing were very large, as shown by the comparison of early, medium and late sowings (Figure 3a–c). Si-c at harvest was reduced from 1975 MJ m^−2^ for the earliest sowing to 208 MJ m^−2^ for the latest.

The earliest spring crop was sown on 28 March or day 86 and later crops over a period of 50 days to day 136 in mid-May. Delay subjected crops to increasing temperature, from 8.4 °C for the first crop to 13.1 °C for the last sown, a rise of +4.7 °C in total or +0.095 °C d^−1^ (Figure 2a). Solar income during the early phases of growth was much higher than during the equivalent period for winter crops, nearing its maximum soon after the first spring crops were sown, then declining with later sowings. A comparable analysis to that for winter crops showed that the solar income received by spring crops in the 100 d after the first sown crop fell from 1490 to 975 MJ m^−2^, equivalent to a decline of 10.4 MJ m^−2^ for each day’s delay after the first sown (Figure 2b). Delayed spring-sown crops developed more rapidly at the higher temperature, but had a shorter duration to maturity, and experienced decreasing solar radiation and a risk of limitation by factors such as drying soil (not quantified in this study). In consequence, the combination of increasing temperature and decreasing solar income in delayed sowings reduced Si-c from 1391 MJ m^−2^ at the earliest to 129 MJ m^−2^ latest (Figure 3d–f). The last-sown crops at both winter and spring sites would be considered in commercial agriculture to have ‘failed’, but are retained in the analysis to show the resulting effects on weeds. Excluding the two ‘failed crops’, the mean with standard error for winter crops was 1576 MJ m^−2^ (±112) and for spring crops 1099 (±121), or 69.7% of the winter.

### 3.2. Relation between Transmitted Radiation and Attributes of the Weed Layer

Transmitted radiation in principle sets the upper limit of radiation intercepted by the weeds, Si-w, but in all examples in Figure 3, Si-w was much lower than St. In winter crops, Si-w hardly increased before the winter due to very low weed cover after sowing, but then began to accumulate in spring from day 50 to 100. In spring crops, Si-w began to accumulate shortly after sowing, from around day 150. Some of the factors that determined low Si-w are revealed in Figure 4. A group of five winter and two spring sites (group x in Figure 4) developed high crop cover and hence low to moderate St (190–610 MJ m^−2^), but the weeds intercepted <10% of this St. All sites in this group were sown at or soon after the earliest date, and were given standard pre-sowing or pre-emergence herbicide treatment (see Materials and Methods). Weed populations at these sites had been rigorously suppressed soon after sowing to the point where they could not take advantage of the little transmitted radiation available. In contrast, three sites (group y in Figure 4) at which weed interception was a higher proportion, 25–30%, of St, were identified as having no or ineffective herbicide application. Two other sites (marked z in Figure 4), one winter and one spring, the latter intercepting 60% of the transmitted, were the latest sowings that produced very low crop cover that allowed weeds to expand almost unchecked (Figure 3c,f). No recorded agronomic influences could be attributed to the three other sites (group w). 

Due to the combined effects in Figure 4, cumulative Si-w from sowing to crop harvest ranged from 14.2 to 734 MJ m^−2^ (52-fold) in winter crops, from 34.4 to 963 MJ m^−2^ (28-fold) in spring crops, resulting in a 68-fold range across all sites. Although effective crop establishment and timing of herbicide treatment were responsible for the lower values in the range, site groups y and z on Figure 4 show the potential for weeds to take advantage of greater transmission when weed control was ineffective. 

Results for total weed matter are presented because the proportions of monocot (11.5%) and dicot (88.5%) mass (Figure 2, inserts) were not systematically related to any of the agronomic variables or solar radiation fluxes (ns, analysis not shown). Total weed dry matter ranged from 6.4 to 208.6 g m^−2^ in winter crops, and from 19.7 to 459.2 g m^−2^ in spring, giving an overall 72-fold range, similar to that for Si-w. Among sites, weed dry matter was closely related to Si-w (F. pr. < 0.001, 93.5% var; equation for winter, y = 0.50x + 14.6; spring, y = 0.62x + 18.5). Because half the sites fall within the lowest 10% of the overall range, the coordinates are displayed on log scales in Figure 5 (with appropriate equations) for visual comparison. The solar radiation use efficiency (Es, Equation (4)) was estimated as dry matter produced per unit of Si-w accumulated up to the weed sample, the means across sites being 0.74 g MJ^−1^ (±0.092) for spring and (the same mean) 0.74 g MJ^−1^ (±0.176) for winter crops. Moreover, Es did not vary in relation to any of the attributes of solar flux (ns, analysis not shown).

### 3.3. Determinants of Weed Species Number

When all species are counted, including those only at cotyledon stage, the number increased from early to delayed sowings: 12 to 27 in winter, and 20 to 40 in spring crops. When only species with established plants were counted (see Materials and Methods for an explanation), the corresponding numbers were 5 to 21 for winter and 12 to 28 for spring crops. These differences in species number were much smaller than the corresponding range of weed mass. Fields supporting very low weed mass of around 10 g m^−2^ in winter crops still had up to half the number of species as fields supporting a weed mass of >200 g m^−2^, and overall, number of species was not significantly related to weed mass (ns, analysis not shown). Species number was also not directly related to weed cover (fw) because, for a similar value of fw during spring and summer, spring crops typically supported up to twice the number of species. Therefore, species number is examined directly in relation to transmitted radiation.

Weed growth occurred mainly in spring (Figure 3), as the incoming solar radiation began its annual rise. The precise timing of weed emergence and establishment was not measured, but preliminary analysis showed that the choice of starting date for accumulating or averaging St in winter crops made little difference to the result obtained, at least partly because of low incoming radiation for much of the winter period. Results are here presented for an arbitrary start at day 0 (1 January) for all winter sites, and day of sowing for all spring sites. The relation between transmitted radiation and species number was highly significant (F. pr. < 0.001) both when the total species count was used (73.8% var), or those species present at >0.1 g m^−2^ dry mass (89.9% var); the latter are shown in Figure 6a. When cumulative St was used (Figure 6a), spring sites extended to more species and had a steeper slope than winter sites. A possible reason for the difference in slope is that emergence and survival in winter crops were happening during a time of the year (early spring and summer) when incoming radiation was lower than in the corresponding period for spring crops. Accordingly, when transmitted radiation was expressed as an average per day, the relations for winter and spring became aligned (Figure 6b), implying that weed species number in both crop types was governed by a similar response to daily transmitted radiation. 

Other weed attributes, including the proportion of dicots (mean 73.4% of species) and monocots (26.6%) and Shannon index were unrelated to transmitted radiation, did not differ between winter and spring crops, and were unrelated to delay in sowing and crop intercepted radiation (ns, regression data not shown). The conclusion for weeds is that transmitted radiation encouraged moderate to low numbers of species to emerge at all sites, but microclimate and control measures at and shortly after sowing determined the subsequent rise of Si-w and weed mass. Species number and mass were to a degree decoupled and weed mass was not restrained by the number of species emerged.

The increase in species number at high values of St occurred mainly due to the addition of common dicotyledonous weeds that typically support the arable food web [6]. Of weeds considered today to be very competitive, wild oat (*Avena fatua*, *A. sterilis*) and cleavers (*Galium aparine*) were found at a few sites only and invariably at low mass. None of the highly competitive or poisonous species, such as *Chrysanthemum segetum* or *Agrostemma githago*, that caused crop loss or spoilage historically [29,30], were present. Of the five species regulated by law since 1959 [31], *Senecio jacobaea* was absent, whereas species of *Rumex* (docks) and *Cirsium* (thistles) were present at few sites at low mass. 

### 3.4. Interaction between Sowing Time, Temperature and Incident Solar Radiation

The sensitivity of weeds and crops in the predominantly winter cropping system to seasonal variation in temperature and solar income is explored systematically using Equation (5). The analysis is based on actual data for temperature, solar radiation and crop cover measured for the earliest sown winter crop. The intention is to quantify the responsiveness of the system to factors such as delay in sowing and a small rise in mean temperature. The time course of fc is derived first by fitting the equation to the values of fc measured before winter and at the maximum in the following summer. The result is shown in Figure 7a by the line marked ‘no delay’. The time course of fc is then used to derive cumulative solar radiation interception by the crop (Si-c), shown in Figure 7b. Cumulative Si-c rises to 1911 MJ m^−2^, similar to the value in Figure 3a. An increase in mean temperature of 1 °C was then applied to daily temperature throughout growth. The increased temperature caused fc to rise more rapidly before winter, reaching between 0.75 and 0.8 (the dashed line marked +1C). Si-c at increased temperature remained above the ‘no delay’ curve, but reached a final value 91% of the former because the duration of growth was shorter (the overall time–temperature integral was reached earlier at a higher temperature). The effect of an arbitrary delay in sowing is shown by a second time course marked ‘20 d delay’ (which was derived using the same daily temperature and solar radiation data as the ‘no delay’ curve). The delay resulted in a much lower fc before winter (Figure 7a). Harvest was assumed to occur within one week of that for the ‘no delay’ crop (as found typically among the trial sites), but the overall effect of delay was to reduce Si-c to 78%. Temperature was then increased by 1 °C with a concomitant rise in fc, which resulted in the final Si-c being raised to just below that of the ‘no delay’ crop at current temperature. The potential consequences for St and weeds of the magnitude of change due to temperature and sowing delay is shown in Figure 7c. Mean St over the period when weed mass and species number were determined (0–190 d) was estimated (as above) for a range of change in temperature between −0.5 and +1.5 °C. Although delay increased mean St at current temperature from 1.8 to 4.1 MJ m^−2^ d^−1^ (comparable with Figure 6b), the effect was countered by an increase in temperature of 1 to 1.5 °C.

## 4. Discussion

Solar radiation flux estimated for weeds may offer a similar unifying approach to that in which crop production is standardised, compared and modelled across different species, locations and climates of the world [46]. Several advantages were gained by the application of solar flux in this study. The methodology enabled crops and weeds to be quantified in the same, and widely used, metrics of solar radiation accumulated per unit area (MJ m^−2^) and radiation use efficiency (g MJ^−1^). The weed layer performed like many crops [23,25] in that the efficiency of conversion was relatively stable while most of the >70-fold variation in mass was attributable to intercepted radiation. The value for weeds of 0.74 g MJ^−1^ measured here was between the values for oilseed rape of 1.35 g MJ^−1^ during vegetative growth and 0.4 g MJ^−1^ during grain filling measured in a broadly similar climate [47].

The methodology revealed relations and effects that could not be quantified using ground cover alone. A given percentage cover does not equate directly to weed mass or number of species because the translation of cover to intercepted or transmitted solar would be modified by the size of the crop canopy, the period over which a particular cover was maintained, and by the intensity and duration of incoming radiation. For example, weed cover in summer was similar for crops of winter and spring oilseed rape, but the latter supports far higher weed mass and species number attributed to the much higher incoming solar radiation during the respective period of weed emergence and growth (Figure 5 and Figure 6). 

The most instructive outcome of the analysis was the chain of effect revealed, in which small differences in agronomic decisions, such as time of sowing, propagated over time through crop canopy expansion to result in wide variation in crop performance and transmitted radiation, with its subsequent strong influence on weed mass and species number. The large downstream effects of delay in sowing were attributed to small differences in temperature after sowing being amplified by rapid change in the annual cycle of incoming solar radiation. Although delay in sowing reduced interception in both winter and spring crops, the link to the climatic variables and the physiological mechanisms operating were different in each case (Figure 2). Propagation and amplification of effects were therefore highly specific to the context. 

At a practical level, the metrics of solar flux used here were developed from non-destructive measures, which can be taken quickly by a field team, rather than destructive sampling of crop and weed matter which is time-consuming and modifies the micro-environment. Moreover, although destructive field sampling was essential for testing the approach, the analysis of solar flux can be accumulated or averaged over any time interval. The methodology may therefore provide an effective tool for monitoring potential future weed problems, given the many uncertainties in the relation between changing climate and weeds [11], and the need to undertake rapid evaluation of unusual weather events [14]. 

### Gauging the Sensitivity of the Crop-Weed Balance

The results also provide insights into two important issues in managing the weed-crop balance, one pertaining to the decline and loss of spring-sown oilseed rape break crops and the other to the sensitivity of winter cropping in future climate. 

The study showed that greater in-field biodiversity resulted when transmitted radiation to the weed layer increased. However, attempts to encourage in-field biodiversity through purposely delaying sowing in winter crops is likely to be too risky, in that adverse weather could shift a short delay to a long one and hence to crop failure. The most secure approach would be to switch back to a spring crop in some years or areas—the result would be twice as many species, but comparatively little increase in weed mass. Other considerations also favour a partial return to spring oilseed crops. The reduction in crop yield to around 70% would be countered by a similar reduction in the amount of nitrogen fertiliser (the largest contributor to greenhouse gas emissions in cropland) from 221 kg ha^−1^ N for winter compared to 153 kg ha^−1^ or 69% for spring crops [40]. In addition, a switch to a spring crop would reduce pesticide applications (fungicide, insecticide and herbicide) from 6.26 applications for winter to 2.25 or 34% for spring crops [41]. Such a switch would be a practical proposition in some years or over parts of the landscape. 

A consequent concern is whether an increase in biodiversity due to less crop interception and more transmitted radiation would encourage a return of poisonous and highly competitive species. None of the trends observed here resulted in an increase in the proportion of grass weeds that are particularly damaging to cereals. None of the poisonous species that had been brought under control over past centuries and decades [29,30,31] appeared in the fields when combinations of sowing time, temperature and incoming solar led to increase in number of species (e.g., in Figure 6). Most of the increases in weed mass and species occurred in the typical broadleaf component of the seedbank. Although species such as *Stellaria media* and *Capsella bursa-pastoris* can at high density reduce yield, they themselves can be controlled in other years by crops such as cereals. The study therefore showed that moving back to a spring-sown broadleaf crop in some years would enhance biodiversity without leading to the build-up of noxious species in the seedbank. 

The broader questions around the sensitivity of the weed–crop balance to climatic variables were also informed by the study. Although agriculture in many parts of the world is being and will continue to be seriously disrupted by climate change [48], crops in the northern part of the UK are likely to be less affected. Analysis of UK weather in recent decades [33] indicates a warming of 0.3 °C per decade since the 1980s and the likelihood of more frequent extreme wet and hot periods. The latest predictions of future climate from the UK government [33], although subject to great uncertainty, indicate that until 2050, winters could become 1 °C warmer and 5% wetter (with a higher frequency of more intense rainfall), whereas summers could be 1.5 °C warmer and 10% drier. A summary of modelling and experiments, mainly in the last two decades, revealed that such changes in temperature are likely to have only small effects on yield of the main crops in the UK: a reduction in yield due to a shortening of crop duration at higher temperature would be countered by more rapid assimilation (in the predominantly C3 crops) due to the rising atmospheric CO_2_ concentration [49]. The small, predicted effects on yield due to rise in temperature in recent decades were later examined and confirmed by modelling yield and weather for the main cereal crop in the region [50]. 

Nevertheless, the results of the present study should indicate where sensitivity may lie in winter oilseed rape after a change in temperature of +1 to +1.5 °C. Very large effects on crop interception and transmission due to both delay in sowing and a rise in mean temperature (achieved by increasing the daily mean by the same amount throughout the year) are manifest during the period of 3 to 4 months between sowing in autumn and the immediate winter. However, crop growth is minimal during this period due to the low and declining incoming radiation. In the simulations in Figure 7, crops that developed more rapidly at a +1 °C higher temperature had a much higher interception over the winter (Figure 7a), but a shorter duration of growth in summer. The result was a small decrease in crop interception by maturity. As argued above, this small decrease would, in reality, tend to be countered by the higher CO_2_ concentration (not accounted for in the analysis here). The rise in temperature would also reverse, to a large degree, the negative effects on crop of a delay in sowing (Figure 7b). It would also lead to less transmitted radiation available to stimulate weed growth and species number the following spring and summer (Figure 7c). Based on evidence here, small increases in temperature are unlikely to enhance weed mass and biodiversity in winter oilseed rape fields. However, the sensitivity of crop interception to weather and agronomy over the coldest part of the year is clearly demonstrated. Whatever the further uncertainties regarding the future climate, for example in rainfall pattern, waterlogging or drought, the very high explanatory power of solar flux supports its application and adoption in monitoring and mitigation. 

## 5. Conclusions

A chain of effect was traced from agronomic choices through the annual cycles of solar radiation and temperature to crop growth, weed growth and arable plant diversity. When sowing date and early weed management were less than optimal, a crop emerged in less conducive weather that curtailed its canopy expansion over subsequent months; it intercepted less solar radiation and transmitted more through the canopy, encouraging more weed mass and species. A large part of this variation was explained through the simple estimates of solar radiation flux, which quantified the causal links between weather, agronomy and plants. The cropping system studied here has shifted in recent decades from a mix of winter and spring crops to predominantly high intensity winter crops. Reinstatement of spring crops as a means of supporting the in-field food web would result in a moderate loss of yield, but also a gain in terms of less fertiliser and pesticide. Although early growth of winter oilseed rape is highly sensitive to the interaction between agronomic decisions, temperature and solar income, the findings on solar flux indicate a small 1 to 1.5 °C rise in temperature would have little effect on the weed–crop balance because of compensatory changes during the following spring and summer. Further data would be needed to assess the effects of possible trends to more frequent and heavier winter rainfall. More generally, the methodology could be adapted to help field management achieve a desired balance of production and supporting ecosystem services over a range of spatial and temporal scales in any part of the world. In situ measurement of percentage cover and the proportions of species and functional traits can be carried out rapidly over large tracts of land, without severe disturbance to the vegetation and, as shown here, can be converted to widely recognised units of cumulative resource interception and use-efficiency. 

## Figures and Tables

**Figure 1 plants-10-02657-f001:**
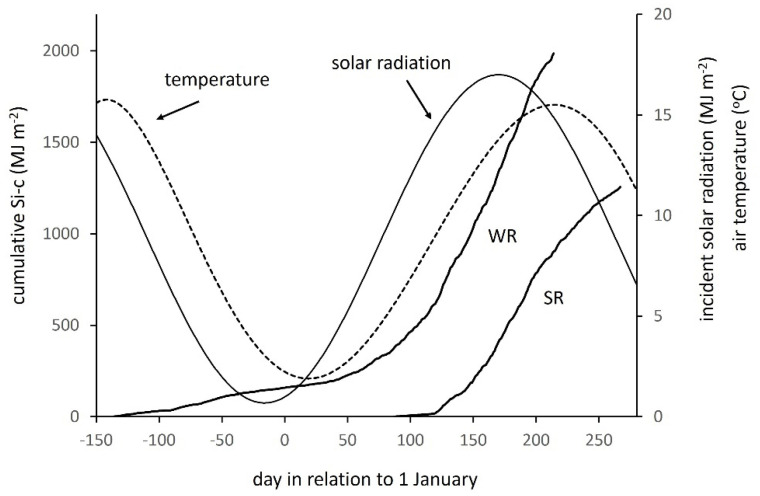
Representative annual waves of incident solar radiation and mean air temperature with growth period of typical winter (WR) and spring (SR) oilseed rape crops shown as cumulative solar radiation interception (Si-c) at Dundee UK, latitude 56.46° N, longitude 2.97° W.

**Figure 2 plants-10-02657-f002:**
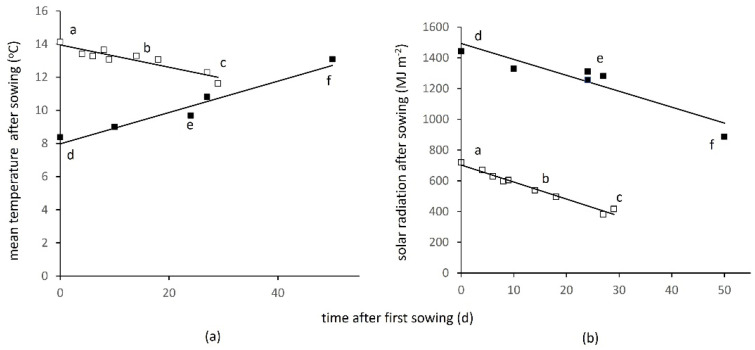
Air temperature (**a**), measured at the meteorological site, averaged over the first 30 d after sowing, and solar radiation accumulated during the early phase of growth (**b**) for winter (open square) and spring oilseed rape (closed square) subject to increasing delay in sowing after the first sown crops: with reference to exemplar sites used in Figure 3, letters a and d were the first sown, b and e intermediate, and c and f last sown.

**Figure 3 plants-10-02657-f003:**
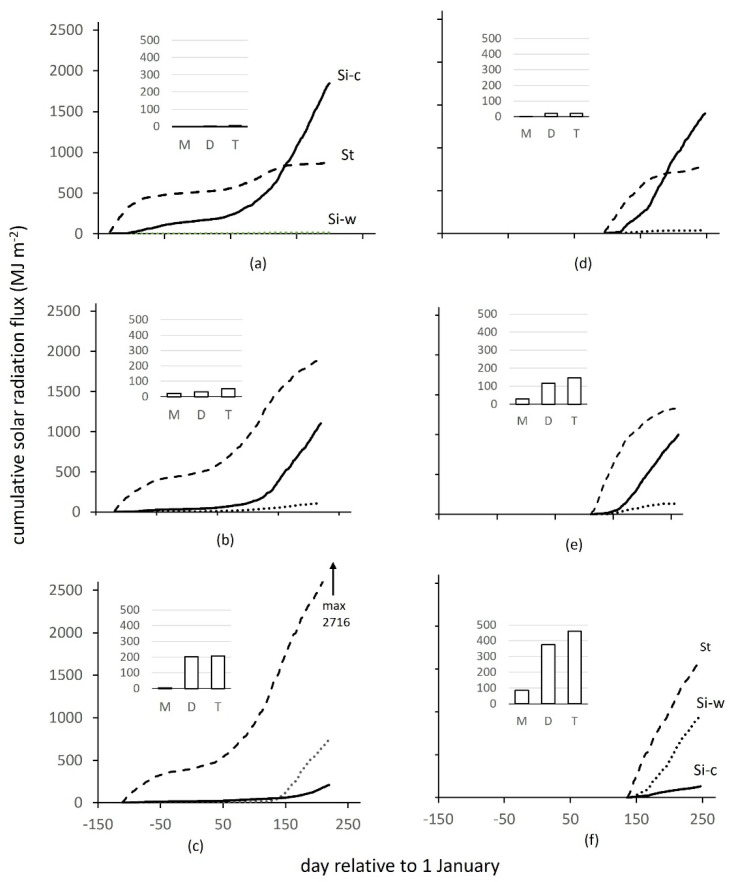
Examples of cumulative solar radiation (units MJ m^−2^) for sites identified in Figure 2 by letters a to f, designated as the earliest sown of winter (**a**) and spring (**d**) crops, intermediate (**b**,**e**) and the latest sown (**c**,**f**), showing solar radiation intercepted by the crop (Si-c, solid line), transmitted by the crop (St, heavy dashed line) and intercepted by the weeds (Si-w, fine dashed), also indicated on the extreme sites to aid interpretation. Inset graphs show weed dry matter (units g m^−2^) for monocot (M), dicot (D) and total (T) species.

**Figure 4 plants-10-02657-f004:**
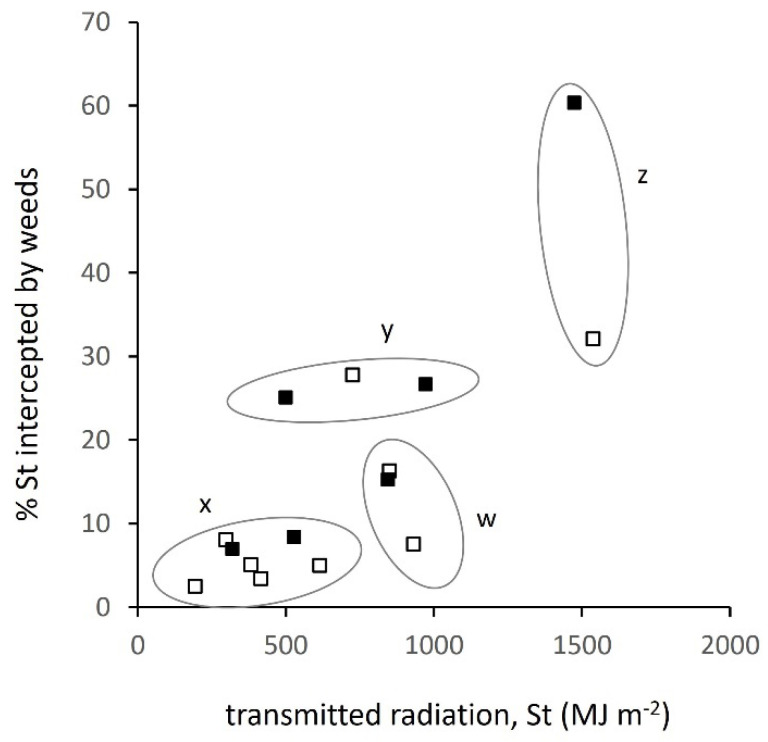
Percentage of transmitted solar radiation (St) intercepted by weeds in winter (open square) and spring (closed square) crops during the main 100-day period of weed growth in spring; site groups, x, early sowing, full herbicide; y, intermediate sowing, poor weed control; z late sowing; w, three sites having higher St but not distinguishable by agronomic characteristics.

**Figure 5 plants-10-02657-f005:**
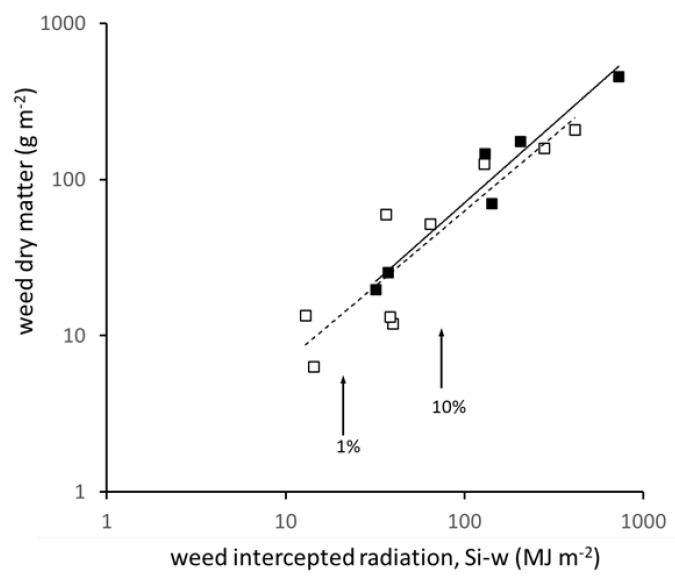
Weed dry matter and cumulative weed intercepted radiation (Si-w) up to the weed sample. Both axes are on a log scale to display the lower part of the range, for winter crops (dashed line, y = 0.74x^0.96^) and spring (solid line, y = 0.67x^1.10^), F. pr. > 0.001, %var 93% (non-log estimates given in text). Arrows show the calculation of Si-w for a weed cover of 1% and 10% between April and August under a mean daily solar income of 15.5 MJ m^−2^ and mean crop cover of 70%.

**Figure 6 plants-10-02657-f006:**
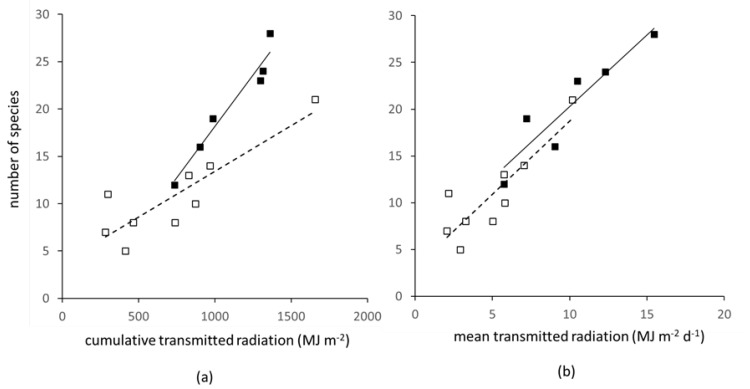
Number of weed species with mass >0.1 g m^−2^ at each site in relation to transmitted solar radiation (St) for spring crops (SR, closed squares) measured from sowing and winter (WR, open squares) measured from day 0, expressed as (**a**) cumulative and (**b**) daily mean. Regression analysis: (**a**) F. pr. < 0.001, 89.9% variation explained, equation for WR is y = 3.41 + 0.0088x, for SR y = −3.59 + 0.022x; (**b**) F. pr. < 0.001, 86.8% variation explained, WR y = 14.5 + 0.51x, and SR y = 4.97 + 1.53x.

**Figure 7 plants-10-02657-f007:**
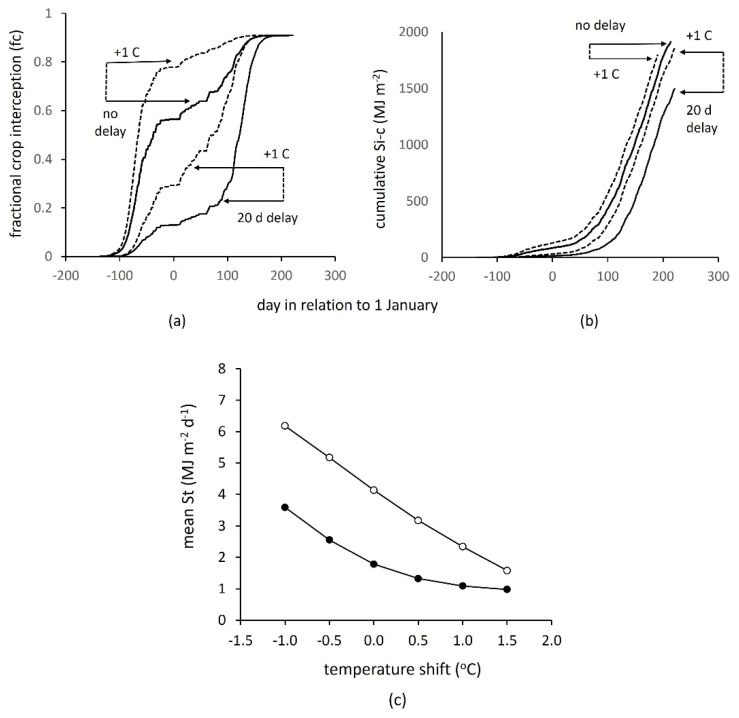
Estimated time courses of (**a**) fractional crop interception or cover and (**b**) cumulative solar energy interception based on conditions of the earliest sown winter crop (no delay) and the same crop subject to a 20-day delay in sowing, and both subject to a rise in daily temperature of 1 °C, with (**c**) resulting effects of temperature shift on mean transmitted radiation (St) during the main phase of weed growth (190 d from 1 January) in the earliest sown crop (closed circle) and the same crop subject to a 20-day delay (open circle).

## Data Availability

The original data from the field trials is lodged at the following: Farm scale evaluations of herbicide tolerant genetically modified crops—spring oilseed rape. NERC Environmental Information Data Centre. https://doi.org/10.5285/0023bd6e-4dd7-462c-aacf-f13083b054ab (accessed on 30 October 2021); and Farm scale evaluations of herbicide tolerant genetically modified crops—winter oilseed rape. NERC Environmental Information Data Centre. https://doi.org/10.5285/37a503da-d75c-4d72-8e8b-b11c2fdc7d92 (accessed on 30 October 2021). The authors will provide species lists per site, met data and daily solar fluxes on request.

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
