# Peer review of "Solar Radiation Flux Provides a Method of Quantifying Weed-Crop Balance in Present and Future Climates"

_plants, 2021, doi:10.3390/plants10122657_

Round 1

Reviewer 1 Report

Plants Manuscript

General comments:

-In general, this manuscript has a valuable topic. The topic is scientifically sound.

-The writing style and English language are fine except for minor English spelling and grammar changes.

experimental design is adequate.                                

-There are some minor comments.

Detailed comments:

-In general, please avoid using personal pronouns such as we, our results, our work and apply this rule throughout the manuscript (for example -Line 23 (we indicate), Line 15 (We evaluated).

Title:

It is suitable, concise, and informative.

Abstract:

This section is missing the direct and clear aim.

Key words: Please take off arable, energy, and change from the keywords list.

  • Please take off intercepted radiation, transmitted radiation and replace them with the word radiation.

Introduction:

The topic is very important and has a great value. I see that the introduction didn’t provide enough background about the topic and needs to be enriched,

Materials and Methods:

The experimental design was suitable and adequate to the current study.

Results:

-I found that the results section is well presented.

Discussion

This section is ok

Conclusion:

The conclusion is supported by the results and includes the most significant findings.

References:

The authors provided enough citations, and it is up to Date.

***This manuscript is very valuable and will be suitable to be published in the Plants journal after minor revision.

Author Response

Reviewer 1

General comments:

-In general, this manuscript has a valuable topic. The topic is scientifically sound.

-The writing style and English language are fine except for minor English spelling and grammar changes.

experimental design is adequate.                                

Authors’ general response: the positive response of Reviewer 1 is appreciated. The authors have made the necessary changes in response to the comments below

-There are some minor comments.

Detailed comments:

-In general, please avoid using personal pronouns such as we, our results, our work and apply this rule throughout the manuscript (for example -Line 23 (we indicate), Line 15 (We evaluated).

Authors: agreed; all pronouns checked throughout the ms.

Title:

It is suitable, concise, and informative.

Abstract:

This section is missing the direct and clear aim.

Authors: abstract revised to emphasise aims

Key words: Please take off arable, energy, and change from the keywords list.

  • Please take off intercepted radiation, transmitted radiation and replace them with the word radiation.

Authors: accepted and changed

Introduction:

The topic is very important and has a great value. I see that the introduction didn’t provide enough background about the topic and needs to be enriched,

Authors: the aim was to keep the Introduction brief as requested by the journal, but your comment above and those from other reviewers have recommended expanding the Introduction which has now been done with additional explanation and references.

Materials and Methods:

The experimental design was suitable and adequate to the current study.

Results:

-I found that the results section is well presented.

Discussion

This section is ok

Conclusion:

The conclusion is supported by the results and includes the most significant findings.

References:

The authors provided enough citations, and it is up to Date.

***This manuscript is very valuable and will be suitable to be published in the Plants journal after minor revision.

Reviewer 2 Report

General comments

Review plants-1465226 – “Solar radiation flux provides a method of quantifying weed-crop balance in present and future climates” by Geoffrey R Squire et al.

The authors have done a lot of interesting work. The Authors presented original research. The study states how the research fills the identified knowledge gap.

Unfortunately, from a scientific point of view, I have a few concerns with this paper.

The Introduction is very general. I miss information about the radiation available depending on the place on the globe. What is the situation in the sub-polar regions and at the equator? How much of this radiation reaches, for example, during monsoons or the dry season?. Moreover, it should be remembered that different species have very different light requirements. I believe that all these (very complex) dependencies should be included in the Introduction chapter. Only later in the text (in Material and Methods section) I can read that: “All fields that occurred between latitudes 56o and 58o 102 N in the East Scotland region were examined”. Thus, I suggest that the authors should focus more on the description of the studied region.

I also believe that some figures should be corrected. The description of the figures as well as the description of their scale should be corrected. Moreover, in some cases figures should be completely changed.

 In conclusion, I believe that the manuscript fits into the Journal’s aims and scope, and I think it is interesting enough to be published after a major revision. 

Specific comments:

L6-7: please check the journal requirements, but it seems to me that you do not need to repeat the same affiliation three times

L19, 32, 40, 100, 212, 229, 235, 245, 254, 273, 288, 470, 473: please remove the double space

L68: please delete the dot

L94: I suggest checking the journal's requirements. It seems to me that the Material and Methods section should be at the end of the manuscript. Then the order of citations should be changed. But please verify it before making any changes.

L98: I think the font here is different than the rest of the text.

Fig. 1: please add the place of measurement to this description.

Fig. 2: I suggest adding the meaning of individual letters to the description instead of redirecting the reader to Fig. 3.

Fig. 3: please make another figure instead of pasting the graphs inside the graphs.

Fig. 3: please add a description of the y scale.

L352: please add a citation, instead of redirecting the reader to the Introduction.

Author Response

Reviewer 2

Review plants-1465226 – “Solar radiation flux provides a method of quantifying weed-crop balance in present and future climates” by Geoffrey R Squire et al.

Authors’ general response: The authors thank Reviewer 2 for their constructive comments and attention to textual detail. The authors have considered each point and responded below and made alterations to the text.

The authors have done a lot of interesting work. The Authors presented original research. The study states how the research fills the identified knowledge gap.

 Unfortunately, from a scientific point of view, I have a few concerns with this paper.

 The Introduction is very general. I miss information about the radiation available depending on the place on the globe. What is the situation in the sub-polar regions and at the equator? How much of this radiation reaches, for example, during monsoons or the dry season?. Moreover, it should be remembered that different species have very different light requirements. I believe that all these (very complex) dependencies should be included in the Introduction chapter. Only later in the text (in Material and Methods section) I can read that: “All fields that occurred between latitudes 56o and 58o 102 N in the East Scotland region were examined”. Thus, I suggest that the authors should focus more on the description of the studied region.

Authors: This point is taken. Some text is added on the global picture and the argument directed earlier in the Introduction to focus on the study region.   

I also believe that some figures should be corrected. The description of the figures as well as the description of their scale should be corrected. Moreover, in some cases figures should be completely changed.

 In conclusion, I believe that the manuscript fits into the Journal’s aims and scope, and I think it is interesting enough to be published after a major revision. 

Specific comments:

L6-7: please check the journal requirements, but it seems to me that you do not need to repeat the same affiliation three times

Authors: guidance has been requested from the managing editor.

 L19, 32, 40, 100, 212, 229, 235, 245, 254, 273, 288, 470, 473: please remove the double space

Authors: changes made.

L68: please delete the dot

Authors: change made.

L94: I suggest checking the journal's requirements. It seems to me that the Material and Methods section should be at the end of the manuscript. Then the order of citations should be changed. But please verify it before making any changes.

Authors: guidance has been requested from the managing editor.

L98: I think the font here is different than the rest of the text.

Authors: checked, but in the ms downloaded the font is the same on as the previous and next lines.

Fig. 1: please add the place of measurement to this description.

Authors: name of place and latitude-longitude given.

Fig. 2: I suggest adding the meaning of individual letters to the description instead of redirecting the reader to Fig. 3.

Authors: change made.

Fig. 3: please make another figure instead of pasting the graphs inside the graphs.

Authors: we have looked at this point, but consider that another figure might make the comparisons less easy to understand since the reader would be asked to cross-reference two -six-part figures.  

Fig. 3: please add a description of the y scale.

Authors: our omission, description added.

L352: please add a citation, instead of redirecting the reader to the Introduction.

Authors: citation added.

Reviewer 3 Report

This manuscript took some data collected nearly 20 years ago for another purpose and attempted to use this data to show that predicting solar flux that was present in these crops could be used to make comments about weed ingress.  The authors then attempt to go further with these correlations to make comments on likely effects of climate change on weeds in crops.

I was confused initially by their term “transmitted by the crop.”  But on Line 203, they clarified that by saying it was solar radiation not intercepted by the crop.  This would be a much better term to use throughout the paper.  Saying that light is transmitted by the crop gives the impression they have taken account of light that passed through the leaves of the plants to give dappled light below the crop plant, which they could not have measured if there were no light sensors placed under crop plants.  They were simply talking about light that didn’t hit crop plants, and no account has been made of light that managed to get through leaves to the soil underneath, and species such as Stellaria media can often survive in this lower light intensity.

There appeared to be excitement over the high correlation between their estimated interception of solar flux by weeds and weed dry matter.  But this is not surprising, as the interception of solar flux by weeds was estimated by using leaf area measurements.  So really what was being reported was a high correlation between leaf area measurements of weeds and weed dry matter, something that has been well established in the literature. 

One of my concerns was that there was quite a bit of variability between how crops were managed.  Some had herbicide applied at the correct time, in which case weeds present were either weed species tolerant of the selective herbicide used or weeds that had established late in the season once the residual herbicide was no longer active.  Other crops had the herbicide applied too late to kill weeds that established at the same time as the crop.  Other crops had no herbicides applied at all.  There was little information on the herbicides used, apart from one reference to metazachlor on Line 121.  Little mention was made that the original trials appeared to involve herbicide tolerant crops, though it is unclear if they were all herbicide-tolerant.  These differences in weed management made it a little difficult to take claims seriously about the effects of solar flux on weed diversity and ingress, because there would be much variability depending on the weed management, which is a point that was made by the authors.  But basically they were using their concept of solar flux to point out what all weed scientists and most farmers already know, that if your crop does not provide good canopy closure fairly rapidly, weeds will establish and become a problem, and they will do this more rapidly if a selective residual herbicide is not applied.  It doesn’t need estimates of solar flux to determine this.

The manuscript then attempted to use this high correlation of percentage ground cover by crops allowing weeds to establish (in the guise of solar flux estimates) to determine whether an increase in temperature in Scotland will lead to more weed problems.  I would have thought that could be best worked out by looking at weed issues in crops in the south of England or in parts of Europe where it is already warmer than Scotland.  Most of what was covered in that discussion was really just discussing general concepts of crop management and weed control, that anything which interferes with rapid canopy closure by crops will lead to weed problems if selective herbicide cannot be used to sort the problem out.  It did not require estimates of solar flux from 20-year-old data to make these points, especially when the data was taken from work that was not designed properly to look at these concepts.  And certainly the data cannot really be expected to be good enough to predict what might happen to weed seed banks over time, apart from the obvious point that if weeds are not controlled properly for successive years, the weed seed banks will get larger and have more diverse species present.

There were a number of typographical errors throughout the paper.  The term “poisonous contaminants” on Lines 63 to 64 was presumably a reference to use of herbicides.  This was rather inflammatory language to be used in a paper on weed science, especially since it appeared herbicides were used in many of these crops to deal with weed issues, and weed problems arose when herbicide was no longer active or had not been applied.

In Figure 1, there were three solid lines, not one, and it needed to be made clear it was mean air temperature.

Author Response

Authors’ general response: The authors thank Reviewer 3 for their challenging comments on the paper and feel that such criticism has led to greater scrutiny of the approach and improvement of the paper. To respond to the major points of criticism, the authors have added capital letters in brackets (e.g. A, B) in the reviewer’s text, then responded in detail to each point at the end of the text, indicating also where changes have been made in the revised version.

Reviewer’s comments:

This manuscript took some data collected nearly 20 years ago for another purpose and attempted to use this data to show that predicting solar flux that was present in these crops could be used to make comments about weed ingress. (A)  The authors then attempt to go further with these correlations to make comments on likely effects of climate change on weeds in crops.

I was confused initially by their term “transmitted by the crop.”  But on Line 203, they clarified that by saying it was solar radiation not intercepted by the crop.  This would be a much better term to use throughout the paper.  Saying that light is transmitted by the crop gives the impression they have taken account of light that passed through the leaves of the plants to give dappled light below the crop plant, which they could not have measured if there were no light sensors placed under crop plants.  They were simply talking about light that didn’t hit crop plants, and no account has been made of light that managed to get through leaves to the soil underneath, and species such as Stellaria media can often survive in this lower light intensity. (B)

There appeared to be excitement over the high correlation between their estimated interception of solar flux by weeds and weed dry matter.  But this is not surprising, as the interception of solar flux by weeds was estimated by using leaf area measurements. So really what was being reported was a high correlation between leaf area measurements of weeds and weed dry matter, something that has been well established in the literature. (C)

One of my concerns was that there was quite a bit of variability between how crops were managed.  Some had herbicide applied at the correct time, in which case weeds present were either weed species tolerant of the selective herbicide used or weeds that had established late in the season once the residual herbicide was no longer active.  Other crops had the herbicide applied too late to kill weeds that established at the same time as the crop.  Other crops had no herbicides applied at all.  There was little information on the herbicides used, apart from one reference to metazachlor on Line 121.  (D) Little mention was made that the original trials appeared to involve herbicide tolerant crops, though it is unclear if they were all herbicide-tolerant.  These differences in weed management made it a little difficult to take claims seriously about the effects of solar flux on weed diversity and ingress, because there would be much variability depending on the weed management, which is a point that was made by the authors. (D) But basically they were using their concept of solar flux to point out what all weed scientists and most farmers already know, that if your crop does not provide good canopy closure fairly rapidly, weeds will establish and become a problem, and they will do this more rapidly if a selective residual herbicide is not applied.  It doesn’t need estimates of solar flux to determine this. (C)

The manuscript then attempted to use this high correlation of percentage ground cover by crops allowing weeds to establish (in the guise of solar flux estimates) to determine whether an increase in temperature in Scotland will lead to more weed problems.  I would have thought that could be best worked out by looking at weed issues in crops in the south of England or in parts of Europe where it is already warmer than Scotland. (E) Most of what was covered in that discussion was really just discussing general concepts of crop management and weed control, that anything which interferes with rapid canopy closure by crops will lead to weed problems if selective herbicide cannot be used to sort the problem out.  It did not require estimates of solar flux from 20-year-old data to make these points, especially when the data was taken from work that was not designed properly to look at these concepts. (A) And certainly the data cannot really be expected to be good enough to predict what might happen to weed seed banks over time, apart from the obvious point that if weeds are not controlled properly for successive years, the weed seed banks will get larger and have more diverse species present.

There were a number of typographical errors throughout the paper.  The term “poisonous contaminants” on Lines 63 to 64 was presumably a reference to use of herbicides.  This was rather inflammatory language to be used in a paper on weed science, especially since it appeared herbicides were used in many of these crops to deal with weed issues, and weed problems arose when herbicide was no longer active or had not been applied.(F)

In Figure 1, there were three solid lines, not one, and it needed to be made clear it was mean air temperature.

Authors: changes made.

Authors’ general responses to the reviewers comments (please refer to letters inserted in the text above).

  • Use of 20-year old data. The reason for using data analysed and presented 15-20 years ago was (as explained in the Introduction and M&M) because such a comparison, that took advantage of the wide range of sowing time from late summer to late spring, could not be made in this region today. As stated, spring sown oilseed rape crops are very rarely sown now – they were phased out shortly after the data were collected in preference to higher yielding winter varieties. The intrinsic value of the original data lies in the crops being grown on farmers’ fields according to commercial practice. It is that realism that this study wished to capture as well as the wide range of sowing times. But more generally, there should be no objection to using archived data. The results of the original experiment were published across many high-ranking journals, constitute a major source of material on the arable weed-crop system and were formally archived intentionally for future use. The data have been interrogated subsequently on several occasions (e.g. Debeljak et al. ref cited) and this current paper is another example of their usage.
  • Transmitted radiation - use and estimation. Transmitted radiation is a widely used term, defined clearly here, which after consideration the authors would prefer to keep. Certainly, solar radiation is modified by many components of a canopy. Measurements using instruments ‘tube’ solarimeters or similar (used many times in the past by the authors) do indeed estimate a spatial average of transmission at different depths, but even using solarimeters to separate interception by different species in line intercrops has to make assumptions in order to partition flux between the components (for example, Marshall, Willey, ref. cited). However, solarimeters are not practicable for large scale monitoring and surveying, which is the scale of operations targeted by this paper. There is analogy here with the wider and prior use of crop cover, which is a simplification, but is a useful and realistic one for the purpose of quantifying resource capture and use efficiency by crops of different species in different climates and soils. Similarly, the paper is examining the proposition that weed cover (though a simplification) can offer similar advantages. In the revision, transmitted radiation is defined by an equation as suggested by another referee.
  • Weed cover and weed mass. The authors accept of course that weed cover is a strong determinant of weed mass. Clearly, weed canopies with very little cover simply cannot capture enough solar radiation to produce much mass. However, the relation between cover and mass is not direct. A stated percentage cover does not equate directly to weed mass or number of species. The effect of weed cover would be modified by the size of the crop canopy, the period over which a particular cover is maintained and by incoming solar radiation. In this study, using cover alone would not allow some important distinctions to be made and quantified. For example, weed cover in late spring and summer is similar for crops of winter and spring oilseed rape, but they are associated with very different weed mass and species number because the incoming solar radiation is quite different when averaged or accumulated over the respective periods of weed growth. Moreover, the analysis of solar radiation flux used here provides quantitative indicators in widely used units of cumulative intercepted radiation and radiation use-efficiency that can be compared across cropping systems and climates. The authors have modified the Discussion to emphasise the value the using solar flux as a comparator.  
  • Conventional crop varieties and management. The authors state they used data from the half-field at each location on which was grown a conventional variety of oilseed rape (typically used by the farmer) and the associated herbicide (metazachlor) which was applied according to the agrochemical company’s ‘label’, i.e. the standard application method for the specific chemical. None of the data presented here were GM or used the associated herbicide. ‘Conventional’ management was chosen because this is what the weed community had been adapted to over previous decades. As is the case in commercial agriculture, weed control sometimes does not work, for various reasons – but again, it was this realism that the authors were hoping to take advantage of and indeed did so. The solar flux analysis allowed the authors to quantify and interpret the differential success by farmers in controlling weeds and hence weed intercepted radiation.
  • Comparisons with current climates elsewhere. The authors agree there is value in comparing possible future climates at their location with a range of current climates elsewhere, for example at lower latitudes. There is a problem with relying solely on this approach, however, because climates are not transferred as a ‘bundle’ – so while current temperature farther south might be similar to future temperature here, other climatic factors will not be similarly transferable. The analysis in this paper showed the very strong interaction between time of sowing and the annual cycles of temperature and incoming solar radiation. In a climatic warming scenario, the temperature here might be similar to that farther south in Europe, but the annual cycle of solar radiation will not be. The authors have extended the analysis to demonstrate this effect.
  • Poisonous contaminants refers not to herbicides but to certain weeds that were prevalent and whose seeds and leaves were once harvested along with crops, leading to poisoning or death of livestock, and in some cases people. The authors do not think that mentioning poison in this way is inflammatory. To make this point clearer (and in response to comments from other reviewers), examples are given of Introduction of which poisonous and highly aggressive weed species have been historically controlled, including those named in various Acts of Parliament, but which no longer pose major problems in modern arable cropping.

Reviewer 4 Report

Overall: The authors apply radiation-based modeling theory for crops to weeds through an understory approach to the overall plant community in agroecosystems. Implications for these considerations are important not only for weed science but intercropping as well. The authors use time-series measurements of groundcover and incoming radiation to calculate cumulative absorbed radiation m-2, which is then used to predict weed dry matter and number of species. The work is conducted across many site-years. The analysis appears suitable but I would recommend the analysis section be compiled entirely in the methods, apart from the field notes and data collection sections. For example: L142 and interpolating the data, calculating accumulated radiation (L144), calculating categories of light interception and transmission (L150 to 159), and the Shannon index (L181). I would recommend clearing defining all the metrics using numbered equations with their parameters clearly defined. I think this would help with clarity. For the discussion, I found the discourse on climate change to be largely unsupported with previous literature and only partly relevant to the modeling considerations over using radiation. What about CO2 changes, C3 vs C4 plants, impacts of seed dormancy?

L49 to 52: Flux is mentioned twice here but I would recommend precisely defining what you mean by flux and resource-flux.

L113: How many site-years are used in the analysis total? Was each of the 15 sites planted 4 times (2000 to 2003 L101)?

L117 to 118: Are these what is referenced to as clusters in Figure 4?

Figure 1: is the x-axis a standard presentation for this unit? If not, I’d recommend changing to day of year perhaps? Also, please define WR and SR in the caption.

L136: interceotion to interception.

L155 to 158: Incoming for crop, transmitted for the weed. This seems to suggest the weeds are occupying an understory canopy to the crop, in terms of light/space competition?

L175: Can you give more specifics on species abundance for interpretation of “most sites”? Perhaps a percentage of each? Were they equally prevalent or was one particularly invasive?

L181: I’d recommend writing the Shannon index as a numbered equation. Please provide a reference as well.

Figure 2: I found the x-axis a bit confusing. Would time after first sowing also work?

Figure 3: Cumulative solar radiation flux? L287: Longest delay in sowing?

L265 to 266: I’d recommend moving to discussion then referencing the specific papers and what cropping systems it’s been shown in of agricultural value.

L275: Please put coefficients in methods or reference the methods here.                            

Figure 4: The x-axis looks a bit clipped. Just to be sure, are those negative numbers? What is the clustered fourth group above y? Please consider defining x, y, and z in the caption.

Figure 5: This is cumulative intercepted radiation by weeds?

L333: Crop type? Do you mean cultivar/variety? This was described as only spring and winter rapeseed only. Please specify.

L352: I don’t think it’s necessary to cite the intro here?

L370: “The delay in sowing (Fig. 2) would have little consequence alone.” What do you mean here? Please clarify.

L388: I looked at this reference, while pesticide applications were included, I didn’t see any results on seedbank work that looked at winter vs spring crops?

L407 to 409: References for oilseed rape or weeds responding this way?

L423 to 439: This section requires substantial references to strengthen the argument or it should perhaps be reduced to light considerations only given the modeling.

L447 to 452: Can you provide any references to support those invasive and problematic weeds that are largely controlled in Britain? In the study, you mention shepherds purse, prostrate knotweed, chickweed, annual bluegrass… are these not considered problem weeds?

Author Response

Reviewer 4

Overall: The authors apply radiation-based modeling theory for crops to weeds through an understory approach to the overall plant community in agroecosystems. Implications for these considerations are important not only for weed science but intercropping as well. The authors use time-series measurements of groundcover and incoming radiation to calculate cumulative absorbed radiation m-2, which is then used to predict weed dry matter and number of species. The work is conducted across many site-years. The analysis appears suitable but I would recommend the analysis section be compiled entirely in the methods, apart from the field notes and data collection sections. For example: L142 and interpolating the data, calculating accumulated radiation (L144), calculating categories of light interception and transmission (L150 to 159), and the Shannon index (L181). I would recommend clearing defining all the metrics using numbered equations with their parameters clearly defined. I think this would help with clarity. For the discussion, I found the discourse on climate change to be largely unsupported with previous literature and only partly relevant to the modeling considerations over using radiation. What about CO2 changes, C3 vs C4 plants, impacts of seed dormancy?

Authors: the probing and constructive comments of Reviewer 4 are much appreciated. Equations are now inserted for the solar flux indicators but the authors are unsure what the reviewers means by ‘recommend the analysis section be compiled entirely in the methods’, since these sections are already in the Methods. The authors would be happy to consider further guidance recommend by Reviewer 4 if this could be clarified. The reviewer’s criticism of the section on climate change in the Discussion is accepted. In response: 1) a short additional section is introduced to Results more explicitly explaining how time of sowing, temperature and solar radiation interact to cause large differences in intercepted radiation, and 2) the emphasis in the Discussion is shifted to show how the methodology could be used to monitor future effects of climate on the weed-crop balance in this region.

L49 to 52: Flux is mentioned twice here but I would recommend precisely defining what you mean by flux and resource-flux.

Authors: changes made.

L113: How many site-years are used in the analysis total? Was each of the 15 sites planted 4 times (2000 to 2003 L101)?

Authors: now clarified in the M&M

L117 to 118: Are these what is referenced to as clusters in Figure 4?

Authors: not directly, to avoid confusion, this passage has been simplified.

Figure 1: is the x-axis a standard presentation for this unit? If not, I’d recommend changing to day of year perhaps? Also, please define WR and SR in the caption.

Authors: the time in relation to 1 January was used to aid reference to comparison of timings between winter and spring crops; for example, a day in late summer or early autumn would have a minus sign before it when referring to (e.g.) time of sowing of a winter variety, but no minus sign when referring to harvest of a spring variety.

L136: interceotion to interception.

Authors: thanks for pointing out the error.

L155 to 158: Incoming for crop, transmitted for the weed. This seems to suggest the weeds are occupying an understory canopy to the crop, in terms of light/space competition?

Authors: in retrospect, there was additional and unnecessary methodology given, which has now been removed.

L175: Can you give more specifics on species abundance for interpretation of “most sites”? Perhaps a percentage of each? Were they equally prevalent or was one particularly invasive?

Authors: seven members of the group of common species referred to were similarly present across sites while several others were common but not found at all sites. The information provided on species abundance has been augmented and transferred to Results.

L181: I’d recommend writing the Shannon index as a numbered equation. Please provide a reference as well.

Authors: the index is referred to only once in the Results so perhaps a separate equation is not needed; however a reference is given.

Figure 2: I found the x-axis a bit confusing. Would time after first sowing also work?

Authors: fine, ‘time after first sowing’ works.

Figure 3: Cumulative solar radiation flux? L287: Longest delay in sowing?

Authors: caption has been checked and modified.

L265 to 266: I’d recommend moving to discussion then referencing the specific papers and what cropping systems it’s been shown in of agricultural value.

Authors: point accepted, change made.

L275: Please put coefficients in methods or reference the methods here.

Author response: explanation now given in the Methods and implications transferred to Discussion. 

Figure 4: The x-axis looks a bit clipped. Just to be sure, are those negative numbers? What is the clustered fourth group above y? Please consider defining x, y, and z in the caption.

Authors: comments appreciated, graph redrawn and re-labelled with caption amended.

Figure 5: This is cumulative intercepted radiation by weeds?

Authors: yes that’s correct.

L333: Crop type? Do you mean cultivar/variety? This was described as only spring and winter rapeseed only. Please specify.

Authors: yes, point clarified.

L352: I don’t think it’s necessary to cite the intro here?

Authors: removed.

L370: “The delay in sowing (Fig. 2) would have little consequence alone.” What do you mean here? Please clarify.

Authors: The sentence was inadvertently clipped. The major consequences of delay in sowing arise from the rapidly changing solar radiation that plants received with delay. The point is clarified in a new section added to Results.

L388: I looked at this reference, while pesticide applications were included, I didn’t see any results on seedbank work that looked at winter vs spring crops?

Authors: the reference given was referred to in Methods as containing links to source data on pesticide and fertiliser (i.e. government yearbooks). To clarify, therefore, that specific reference is removed at that point, and references with links are given in the M&M to the government source data.

L407 to 409: References for oilseed rape or weeds responding this way?

Authors: our omission, text amended and reference now given to a summary paper on the effects of predicted climate change n the region on crops and pests.

L423 to 439: This section requires substantial references to strengthen the argument or it should perhaps be reduced to light considerations only given the modeling.

Authors: criticism accepted; the new section in Results shows the large changes in radiation flux that are likely in predicted climates while the Discussion is shortened at this point, refering mainly back to that new section in Results.

L447 to 452: Can you provide any references to support those invasive and problematic weeds that are largely controlled in Britain? In the study, you mention shepherds purse, prostrate knotweed, chickweed, annual bluegrass… are these not considered problem weeds?

Authors: weeds that have been severely problematic (poisoning harvest, causing widespread crop failure, those banned in the 1959 Weeds Act) are now summarised in the Introduction, while the Results has been expanded to conform such weeds were not found or occurred infrequently at low mass. The 20-40 species typical of current seedbanks can cause problems if allowed to increase in density, but the species in this study (and found typically throughout the region) comprise readily controllable weeds that are also beneficial to the arable food web. These points are clarified in the Discussion.

Round 2

Reviewer 2 Report

General comments

Good work with the revision! My main concerns have been addressed. The authors did a good job with the corrections.

I believe that the manuscript is suitable for publication. Congratulations!

Just before publishing please make sure that numerous editing errors (double spaces, double dots at the end of a sentence etc) will be corrected.

Author Response

Thank you for the acceptance. Some of the spacing errors arose through the many track changes, but we have gone through the revised paper carefully as requested.

Sincerely

G R Squire on behalf of the authors

Reviewer 3 Report

It appears that I am out of step with the other three peer reviewers who had less concerns about the manuscript than I did.  I have read through the various counter-arguments from the authors to the points I made, and have also read the modified manuscript.  I am prepared to accept that what is being put forward is worthy of publication even if I don’t agree entirely with the importance of what is being suggested.  I don’t think this difference of opinion should stop publication, and concede that some good points were made by the authors in response to my comments.  My only concern now is that the manuscript needs tidying up as a number of typographical errors have now crept in, with lots of double spacings in places, a number of repeated full stops, and spelling errors such as incorrect spelling of “below” on Line 132, “oven” on Line 198, “air” on Line 275, “Cirsium” on Line 417, “predominantly” on Line 154, and clumsy grammar on Line 139.

Author Response

The reviewer's decision is noted and appreciated as is the time that have taken to re-read the paper. Various formatting and typographical errors have been corrected.

Sincerely

G R Squire on behalf of the authors.